# Understanding Different Aspects of Caregiving for Individuals with Autism Spectrum Disorders (ASDs) a Narrative Review of the Literature

**DOI:** 10.3390/brainsci10080557

**Published:** 2020-08-14

**Authors:** Hadi Samadi, Sayyed Ali Samadi

**Affiliations:** 1Department of Philosophy, Faculty of Law, Theology and Political Science, Science and Research Branch, Islamic Azad University, Tehran 1477893855, Iran; 2Institute of Nursing Research, University of Ulster, Newtownabbey BT37 0QB, Northern Ireland, UK; s.samadi@ulster.ac.uk

**Keywords:** parental impact, caregiving, families living with ASD, autism spectrum disorders, parental engagement, narrative review, review of reviews

## Abstract

Background: There has been a considerable endeavor to understand associated challenges of caregiving for a child with Autism Spectrum Disorders (ASDs) and to develop the necessary skills and approaches to assist parents of children with ASD. Different studies have been stressed the importance and need for parental involvement in the intervention process to increase positive impacts. Methods: The process of caregiving and the associated challenges should be understood from different aspects to be able to facilitate parent involvement in intervention implementation. In a narrative literature review, ten selected reviews were considered and each review considered a special aspect of caregiving for an individual with ASD. Results: Five main different factors in the available literature and reviews were considered as different themes that needed to be reconsidered in the studies on the impacts of caregiving for an individual with ASD. Conclusions: It is concluded that to facilitate parental involvement in the intervention process, and to support caregivers of this group of individuals this review highlights the need for improved research in some proposed areas in this field and to bridge the gap between research and practice in this field.

## 1. Introduction

Despite the recognition of Autism Spectrum Disorders for more than 75 years, the etiology, prognosis, and short or long term impacts of this diagnosis on caregivers have not yet been fully identified [1]. Present knowledge on ASD indicates that it is a lifelong neurodevelopmental disorder in which genes play a role but that environmental triggers likely contribute as well [2].

This lifelong disorder impacts sociocommunicational ability along with aspects of behavioral differences manifests itself through restricted interests or unusual behavioral repertoires [3].

The dominant contemporary idea is that children with disabilities such as children with ASD should not be separated from their parents [4] and families should play a more influential role in the treatment process [5]. Families who give care for a member with ASD can be referred to as families living with ASD [6] because generally, this is a lifelong process.

Understanding the impacts and psychological issues of the diagnosis of ASD on parents as the main caregivers have developed markedly in the last three decades [7]. Parents may experience emotional stress, anxiety, fear, and guilt, and based on the effectiveness of child-centered treatments they might simultaneously show some positive feelings [8].

There are only in some affluent countries and on some special occasions families given the opportunity of respite care or foster home or residential house [9]. Respite care is a break that parents in the affluent countries can have access to which involves a few hours/weekends that the child with special needs will be watched by someone else who will get paid by the state, but it is an underused service for several reasons such as difficulty to find a good respite worker. Foster care is a service available in affluent countries like the United States when parents are unable to care for children in their family homes and children have to be removed for a while until they can gain parenting skills or improve their socioeconomic status. Residential care (otherwise known as institutionalization) is mostly phased out in most states in the USA as disability advocates are moving towards in-home care with supports. Institutional care of children and adults with intellectual or developmental disabilities are more common in non-affluent countries. Therefore, it can be concluded that nearly all children with ASD like the rest of the children with other types of disabilities live at home and with their family members.

To develop and boost skills and approaches to assist parents of children with ASD considerable endeavor has been done. Parental involvement in the intervention process has been stressed in different studies. Involving parents in treatment implementation is advisable but to be able to facilitate this involvement different aspects of caregiving should be understood and taking into account.

This paper reviews key areas of existing literature focused on the parents of individuals with ASD to highlight different aspects of caregiving and the need for understanding the different impacts of caregiving. Therefore based on the Mayer [10] classification of different types of reviews, this paper is a narrative review in which few selected reviews are compared and summarized based on the authors’ experience, existing theories, and models to understand the possible concerns, study trends, and ideas for future studies.

The ultimate goal of this review is to understand the potential of caregivers and caregiving process for an individual with ASD and to understand different aspects of caregiving for individuals with ASD in different societies and to recognize different impacts that caregiving might have on caregivers. This is done to answer the following questions;

Have the presented reviews covered the different aspects of caregiving for an individual with ASD?

How did the presented reviews on caregiving for an individual with ASD echo different theoretical frameworks to explain the phenomena that are considered to be investigated?

How did the presented reviews reveal the geographical distribution of the studies on the impacts of caregiving for an individual with ASD?

Based on the aims of the narrative review, which are described, and discussing the state of the science of a specific topic or theme from a theoretical and contextual point of view, this review also tries to highlight the research trends in the available literature and stresses the areas that are less studied through critical analysis of the literature published in books and electronic or paper-based journal articles.

### 1.1. Caregivers-Focused ASD Research

It has been long identified from previous studies that caregiving for a member with ASD are more at risk from psychiatric and/or stress disorders because of the range of distinct challenges they have [11,12].

The presence of a child with ASD seriously affects the family system as a whole [13,14]. It may have both negative and positive consequences for parents [15]. The majority of individuals with ASD require assistance with their daily routine activities, which is mainly provided by the caregivers who are family members. The activities are in a wide range and cover areas such as self-care, mobility, communication, and cognitive or emotional demands [16,17]. This is why many caregivers of individuals with ASD experience challenges with their general health compared to those who are caring for typically developing individuals or those who have other types of developmental disabilities [18,19,20,21].

### 1.2. Quality of Life and Sources of Stress among Family Members and Caregivers of Individuals with ASD.

To identify challenges associated with caregiving to a member with ASD in family settings many studies have been done and they mainly focused on the qualities of life and maternal stress as mothers are generally the main caregiver for individuals with ASD in the family.

Different factors are contributing to challenges associated with the caregivers’ quality of life and general wellbeing. The level of functional impairment and the presence of challenging behavior appear to contribute specifically to parenting challenges [22]. For example sleep problems of children with ASD could be considered as a source of parental stress. The most common problems reported were “bedtime resistance” and parasomnias [23], which refer to any sleep disorder such as sleepwalking, teeth grinding, night terrors, rhythmic movement disorder, and restless legs syndrome [24].

To understand the contributing factors of challenges for parents of children with ASD investigation on other factors needs. Contributing factors might be coping strategies, available formal and informal supports, satisfaction with caregiving, and family functioning. It is concluded in many present studies that greater social supports for the caregivers, obtaining adaptive coping strategies and caring for an individual with a milder form of ASD related behaviors tend to adjust more easily to the caregiving demands and experience lower levels of stress [25,26,27,28,29]. The present findings extended the existing knowledge and instead of finding the sources of the distress of ASD caregivers inside the family system, we adopted a broader view and discovered that parents of this group of children were motivated to continue employment because of the extra expenses caregiving imposed on them [30], but there are serious burdens on the way of their employment such as a lack of proper childcare services and inflexible employment situations [31]. As a new threat to the general wellbeing and level of stress of caregivers of an individual with ASD unemployment has been added to the list [32].

The key phrases for this paper are parents of children with ASD and the care that they are providing to their child with ASD. To operationalize this phrase it should be defined as the caregiving and supports characterized by attention to the needs of their child; particularly for those unable to look after themselves sufficiently due to the diagnosis of ASD and involved in the provision of their health or social care. The focus of this narrative review is on the presented reviews on parental caregiving regardless of their age level and it covered both young and old caregivers.

## 2. Materials and Methods

In recruiting the review papers for this review, review papers published from 2010 to 2020 were considered, Google Scholar as the main base and publicly accessible source considered. The main focus was on recent review studies published in peer-reviewed journals to investigate the impact of ASD on parents in English language journal papers. The “Autism parents Review”, “Autism caregivers’ impacts review”, and “family impacts and Autism review” were used as the keywords. The titles of the recruited studies are mentioned in the first column of Table 1. Several other keywords such as “parents”, “family members”, “caregivers”, “mothers”, and “fathers” also were used along with the backbone word of “review” to find desired review papers. The results are depicted in Table 1. Ten review papers in all were considered for this review.

The main purpose of this review was to find out what researchers have in mind when they review the impact of caregiving for an individual with ASD.

## 3. Results

Out of ten reviewed reviews, four of them considered the theoretical framework (TF) as one main factor in their review and some others reviews although TF was named or mentioned hence, it was not considered in the review as a factor. Although the wealth of data out of the considerable amount of studies reviewed it was mentioned that studies mainly recruited mothers and fathers are rarely investigated. The recruited sample of fathers in those studies that considered both parents was far lower than the number of mothers. Available studies rarely focused on the positive side of caregiving. Additionally, only one review (in Singapore) was done in non-western societies and considered the cultural factor as a factor to consider in the review. There was only one review that considered the impacts of the level of functioning of ASD as a factor. The rest of the other reviews although tried to consider the level of the functioning or severity of the symptoms in some of the recruited studies, but mainly considered ASD as a general diagnostic term regardless of the level of functioning based on the formal diagnostic procedure.

The following part based on the narrative review aims to find important topics on different general findings of the reviewed reviews presented under the different extracted subtitles. Therefore this narrative review of the reviews will give a detailed explanation of the important findings of the recruited reviews.

### 3.1. Impacts of Adopting a Theoretical Framework

The theoretical framework acts as a guide or plan for a study. According to philosophers of science like Thomas Kuhn [43], observations are “theory-laden” and impacted by the theoretical presuppositions considered by the researcher. That is an inevitable part of every research that needs data collection, so degrees of contamination of data by background assumptions are unavoidable. However, the explicit mention of the theoretical work acts as a caveat for the reader. The reviews reviewed for the current paper have been based on a range of theoretical models. In some reviews these theories were explicit but often the theories were not considered as independent under the review factor. Based on the World Health Organization [44] suggestion any theoretical framework that is adopted for studies on impacts on caregiving for individuals with ASD should consider at least the following three criteria:Cultural issues: the theories had to consider the impact of the social context, cultural influences, and attitudes.Compatibility with the family-centered approach: theories had to be compatible with family-centered approaches.Conceptions of disability: the theories needed to reflect modern thinking about disability, such as is reflected in the International Classification of Functioning (ICF).

Set against these considerations, the two most promising frameworks were ecological approaches to the family [45] and family systems theory [46]. Having a review of the present studies also highlights the limitations of basic research designs adopted from the considered theoretical framework in an attempt to understand the complex interplay between contributing factors of the challenges associated with caregiving for individuals with ASD. As an example to depict the impacts of adopting different theoretical frameworks is that as Turnbull, et al. [47] say, while some theoretical frameworks try to “fix” the individual with a disability and having him/her fit into different levels of family, community, and society, the ecological model’s [48] main endeavor is on “fixing” the multiple ecological environments. Therefore, according to an ecological model, the focus is on a transformed ecology in which children with different types of disabilities can develop through the interaction of their skills with a responsive context [47]. Therefore, in this TF “passive change” as an important factor is taken into account in which a person is not entirely passive and can cause changes in his/her context [49]. Another used TF is the Double ABCX model [50,51], which consists of three components: demands, capabilities, and meanings (situational appraisals) and has been employed to understand the psychosocial impact of children’s chronic conditions in parenting and the factors affecting their adjustment to the child’s diagnosis. In sum, the different and changing nature of the under investigation caregiving factors urged the researcher to adopt different TFs. To bridge the existing gap between research and practice, it became evident that there is a need to expand TF through adopted TF conceptual models that are consistent with each other, and could cover conceptual shortcomings and generalize findings [52]

### 3.2. Impacts of Recruiting Convenient Samples

Most of the studies considered mothers and research on the impact of caregiving for a child with different types of developmental disabilities such as ASD on fathers has been infrequent [53,54]. One of the obvious limitations with the present data is that they are skewed towards mothers and in most of the present studies the mothers’ perspective is considered to be the perspective of all the caregivers in the family [55,56]. Although mothers are considered to be the main caregivers in most cultures, different caregivers in the family might have a different experience in the process of caregiving for an individual with ASD at home [57]. Although the word “parents” is in the title of most of the studies in the files of caregivers’ impacts, the research often was undertaken exclusively or mainly on mothers. Nevertheless, recognizing fathers and their needs has received more attention in recent studies [58]. While available reviews suggest that fathers of children with ASD are not often included in research on individuals with ASD, American Counseling Association [59] proposed special counseling for fathers with counselors deeply considering the fathers’ cultural context. Researchers in the field of developmental disabilities have identified fathers with different words and adjectives, words such as “hard to reach” [60], or “just a shadow” [61]. One main justification is that mothers are generally the main caregivers of their children with special needs globally. Bailey and Powell [53] suggest that mothers tend to spend more time with their children with developmental disabilities and they are more available to participate in studies. According to Altiere and Kluge [62], although the experience and behavior of fathers of children with ASD were considered important, it has not been evaluated consistently. A review of the available literature on child and family psychopathology revealed that 48% of the studies assessed mothers exclusively and 1% assessed fathers. Traustadottir [63] believed that this is because in families of children with developmental disabilities mothers are less likely to be employed in paid jobs and they are expected to take the major caring responsibilities for the child.

### 3.3. Focusing on Negative Aspects Of Caregiving

In one of the reviews it mentioned that only a small number of parents highlighted the positive impacts of caregiving for a child with ASD on parents/caregivers [35]. There is substantial evidence that the presence of a child with ASD seriously affects the family system as a whole with both negative and positive impacts [14]. There is a considerable amount of studies on the stress and wellbeing of families who have a child with ASD [64,65]. However, only a small number of the available existing research also recognizes some positive influences of ASD on caregivers and their overall functioning, including psychological and emotional strength, improved communication skills, and higher levels of empathy and patience [66] reported positive impacts associated with bringing up a child with ASD such as increased spirituality or increased compassion and acceptance of differences reported by Pakenham, et al., [67] or Hastings and Taunt [68] found that positive perception of parents of children with different severe forms of disabilities such as ASD could help parents to cope with high levels of stress and serve as an adaptive function.

In a review of the available literature on the impact of caregiving for a child with disabilities, which has been done by Savage and Bailey [69] in Australia, they found that generally researchers found less satisfaction with life and caring among parents of this group of children. They also found some other studies about the positive impact of caring of a child with disabilities on parents, in which factors such as giving pleasure to the care recipient, maintaining the dignity and maximizing the potential of the care recipient were mentioned by parents [68].

### 3.4. The Dearth of Cross-Cultural Studies

Internationally, most studies were done in the west research on parents of children with ASD and the effects of having a child with ASD to date have been limited largely to families in western countries, and there is a dearth of studies in non-western counties [70]. Bailey and Powell [51] reported that different cultures have different opinions about ASD. Nonetheless, in addition to the World Autism Organization, national organizations for children and families with autism now exist in over 80 countries, suggesting that at least the diagnostic category has traveled around the world. Hence even inside multicultural countries, there are tendencies toward focusing on research interest towards special groups such as White Euro-American families [71]. Unfortunately, there is a dearth of studies regarding the impact of the condition on parents of children with ASD in less affluent countries.

In sum, although there is an appreciable amount of literature on families of children with different types of disabilities, including ASD, in western countries, little is known about the experience of parents in non-western societies. A small group of the reviewed studies investigated the impacts of cross-cultural factors on caregiving for an individual with ASD as a process in different contexts. Due to the possible impact of cultural factors on parental adjustment to the demands of this special type of caregiving, it is recommended that future studies consider the effects of cultural values through adopting cross-cultural studies.

### 3.5. Considering ASD as a Single Diagnosis with Similar Impacts

Individuals with ASD are a very heterogeneous group with different types of abilities and challenges and some studies attributed this heterogeneity to different factors such as genetic heterogeneity among the members of this group [72]. A hallmark of ASD and different needs is due to heterogeneity in etiology, phenotype, and outcome. Different support and services and a variety of impacts on caregivers might be an inevitable output of this heterogeneity. Hence, most of the reviewed studies combine different levels of ASD into one general class of diagnosis. Some studies stress the need to investigate a different type of ASD separately. As an example, the impacts on caregiving for the high functioning group are less studied [73] and most of the available considered the severe forms of ASDs. In a concise review of the literature, it was concluded that understanding of ASD subgroups, their associated markers of pathological states, and different cross-cultural factors such as impacts on the family are imperative to advancing this field of research [74].

## 4. Discussion

There is a growing interest in studying the impact of caregiving for children with ASD on parents. Hence, there are aspects of the available literature that needs revision.

It cannot be neglected that there is a need for studies that reflect on the bigger picture and create linkages rather than perpetuating the highly specific compartments in which much of the present knowledge and understanding about ASD is imperfectly created.

Instead of reviewing the studies, we believed that highlighting the present concern in the available literature as an aim might be better achieved by undertaking a narrative review of the various literature reviews that have been published in the past decade through a ‘review of reviews’ that is rarely undertaken and they lend themselves well to a narrative review.

We believe that this type of review would provide a stronger basis to identify the different levels of theoretical frameworks; for example, a high-level TF such as ecological approaches provides an over-arching framework in which other frameworks such as family systems theory or stress/coping could be defined.

TF as a guiding base for the research in this field was considered as one of the main factors in this review. The theory is defined as an expression of knowledge, a creative and rigorous structuring of ideas that project a tentative, purposeful, and systematic view of phenomena [75]. As Waterhouse [76] suggests, without adopting an appropriate theoretical framework, studies are more likely to be influenced by extraneous factors such as social consensus, convenience samples, opportunities for immediate applications, and researcher preferences. To limit the probability of these risks there is a need for adopting a generic theoretical framework. The reason for considering the common theoretical frameworks in any scientific field is to provide an explanation for the connections among the phenomena under investigation and to provide insights to discover new relationships between phenomena [77]. Swanson [78] concluded that the benefits of a theoretically driven body of work include the utilization of common terminology to improve communication of findings, research methodologies grounded in theoretically sound concepts, and a greater synthesis of results from various individual research studies allowing for detection of emerging patterns.

The review also revealed ambivalence between different negative and positive sides of caregiving. Parents who have a child with disabilities are not automatically under stress [79]. Some studies [66] found that some families have been able to cope successfully and control stress conditions. In other words, there are several studies on the positive impact of having a family member with a disability on their quality of life and strengthening of the family members as a unit [80,81]. Having a member with ASD in the family may have both negative and positive consequences for parents [82] For example, concepts such as “benefit finding” and “sense-making” was explored as the perception of parents of children with ASD [67]. They found that there are parents who are trying to understand the way that their children with ASD perceive the world around them and this endeavor might improve the parent–child relationship. This was considered a positive consequence of having a child with ASD. This aspect might be worth exploring further with parental caregiving to a child with ASD globally. For example, many caregivers report various positive psychological outcomes attributed to parenting their offspring with ASD including selflessness, compassion, and peace during hardship such as a time of uncertainty and a refocus of energy [40,66]. Research into the positive impacts of ASD on families is encouraging but is only relatively recent. Additionally, there are areas such as parental resilience, traumatic growth, family relationship, different aspects of development, and appreciation of life and enrichment of relationships that would benefit further research [83,84,85]. Positive psychology has already recommended similar approaches within the field of developmental disabilities research [66,67,68]. At present, there are mixed results and opposite findings with some reporting positive effects in areas such as self-concept and self-competence of the family members [86,87] and mainly negative impacts on areas such as stigma mainly in the form of social embarrassment [88,89] and psychological distress [90]. On the other hand, some findings indicated no differential impact in areas such as self-concept, self-efficacy, and locus of control [91,92]. The presented work is justified differently, hence, it reflects different impacts of caregiving on caregivers. The findings also revealed the dynamic nature of caregiving, which prohibits any “cause–effect” and direct simple relationship between the under investigating phoneme [57]. Furthermore, the mixed findings of the studies might be an indicator of the attributing factors that are not investigated, understood, or taken into account in previously implemented research designs; factors such as level and sources of parents and caregivers information, family type (extended or non-extended), caregiving at different stages of life, and a range of cultural and demographic factors such as socioeconomic status, nationality, and locality and the severity of a specific diagnosis such as ASD [40]. It is concluded that present inconsistencies in the interpretation of the findings of the available wealth of data are due to considering the contributing factors in isolation and away from other possibly related causative issues [78,93,94].

It is also an important factor to conclude this review with, although ASD is considered as a global public health concern [95], it is estimated that approximately 90% of individuals with ASD live in low/middle-income countries [96]. Hence most of the studies are done in high-income countries and the resulting data due to a significantly different situation may result in very different consequences. There is a growing urge for doing cross-cultural studies in this field because culturally adapted parental services for ethnic minorities could also contribute to the diversity of the parental support and training services in the countries in which immigrants live. This is also relevant to high-income countries that admit immigrants from different cultures, to offer more culturally sensitive services concerning the supports provided to parents as caregivers. The main extracted findings from the reviewed reviews in this paper are not unique to the level of the development of the countries and there are particular challenges for parents of these children, which is global and of relevance to different nations. Furthermore, additional cross-cultural research, albeit within a local context in different countries, is essential if the international understanding of ASD is to be boosted globally.

In sum, it is reported that impacts of caregiving for an individual with ASD is multifaceted and pervasive [97] the main reason for this justification is that approximately 85% of individuals with ASD present with different types limitations and disabilities such as cognitive and/or adaptive in a degree that reduced their possibility of living independently. This lifelong condition caused livelong supervision or assistance in different degrees from their parents or a family member as the main caregivers [98]. Longitudinal studies revealed that almost 50% of over fifty parents of an individual with ASD indicated that they are still caregiving for their offspring with ASD [99]. Caregiving of a child with ASD might have different impacts on different family members and all members deserve to be taken into account in the studies on impacts of caregiving. Parents and caregivers should receive adequate attention and services. It seems that providing opportunities to both parents in a balanced way and considering the ideas and impacts of ASD on both parents and also giving the opportunity of hearing their voices through their own words may produce more generalizable and reliable results.

Not only parents of children with ASD consisting of a diverse group of people with different backgrounds and needs, but individuals with ASD also are a very heterogeneous group with different levels of abilities or functioning that makes it difficult to consider an individual with this diagnosis as an equal group with similar impacts on caregivers. Underrating the level of functioning and severity of symptoms might yield less trustful results. Proposed changes to the DSM-5 in 2013 include dimensional assessments intended to allow clinicians to rate both the presence and severity of psychiatric and related symptoms in a clear way within diagnostic categories [100]. The proposed revisions about the diagnosis of an Autism Spectrum Disorder (ASD) include a severity marker based on the degree of impairment in the domains of social communication and restricted and repetitive behaviors as the dyad of impaired core symptoms. The most recent revision of the Autism Diagnostic Observation Scale—Second Edition (ADOS-2) [101] provides guidelines for calculating the overall level of autism symptoms relative to individuals with ASD of the same age and language level using a rubric called Comparison Scores (CS).

Another possible source for classification following the level of abilities is the International Classification of Functioning, Disability, and Health (ICF), which is a classification and description of functioning, disability, and health using a biopsychosocial theoretical concept that classifies information into four components to classify individuals with a different type of disabilities such as ASD. These components are (1) body functions and body structures, (2) activities and participation, (3) environmental factors, and (4) personal factors. These factors interact with each other to influence the functioning and are classified and described in the ICF manual [102]. These systems might be able to be used for classifying the level of functioning and degree of severities of individuals with ASD.

Although considering that the methodology and data analysis approaches were out of the coverage of this review, most of the available studies adopted a quantitative approach and the mixed approaches or qualitative method is rarely used. Most of the studies adopted a voluntary survey approach in which sampling bias is more possible to happen. This group of participants is more active and open to share their experiences and they do not necessarily echo the existing ideas of all caregivers of the ASD population.

The role of the cultural components can be considered in qualitative studies in more depth. Participants of the qualitative or mixed approaches studies get the opportunity of expressing their ideas through their own words.

## 5. Conclusions

The present review of the reviews highlighted a lack of strong empirical evidence on the structure of studies that considered impacts of caregiving for an individual with ASD on parents/caregivers.

ASD implies a very heterogeneous group of individuals and this is a multifaceted diagnosis with a range of severity and levels of abilities. Individuals place a range of demands on their caregivers, and parents have varying challenges that need to be responded to reduce any additional pressures associated with caregiving. To be able to prepare caregivers of individuals with ASD different aspects of caregiving should be understood.

This review also cautions against the acceptance of the impacts of caregiving at face value and recommends that at the beginning the context and structure of caregiving should be established, before determining links between different aspects of under investigated contributing factors.

In this note, future evaluations need to be done to understand different aspects of caregiving in different cultural contexts to facilitate issues such as parental presence in the intervention process and to provide effective parental support and services packages.

## Figures and Tables

**Table 1 brainsci-10-00557-t001:** Reviews about impacts of caregiving on parents of children with an Autism Spectrum Disorder (ASD).

The Review Title	Author(s)	Number of Reviewed Studies	Main Findings	The Geographical Area That the Study is Done	Considering Autism as a General Diagnosis/with Subtypes	Theoretical Framework Considered in the Review
The quality of life of parents of children with autism spectrum disorder: A systematic review	Vasilopoulou, and Nisbet (2016) [33]	88 studies	Compared to parents of typically developing children or population norms, parents of children with ASD show a poorer quality of life. Contributing factors of parental quality of life were discovered to be the behavioral challenge of the child with ASD, parental unemployment, mother caregivers, and lack of social support for parents.	UK	Autism Spectrum Disorders (ASD) considered as one main diagnosis	TF in the reviewed studies was not considered
Mindfulness, Stress, and Well-Being in Parents of Children with Autism Spectrum Disorder: A Systematic Review	Cachia, Anderson, Moore (2010) [34]	10 Studies	Reviewing the efficacy of interventions in reducing stress and increasing parental psychological wellbeing indicates that all included studies contributed to the efficacy of mindfulness interventions in reducing stress and increasing parental self-reported psychological wellbeing	Australia	Autism as a general diagnosis is mentioned	TF in the reviewed studies was not considered
Couple relationships among parents of children and adolescents with Autism Spectrum Disorder: Findings from a scoping review of the literature	Saini, M., Stoddart, K.P., Gibson, M., Morris, R., Barrett, D., Muskat, B. and Zwaigenbaum, L. (2015) [35]	59 studies	Factors that support the development and maintenance of positive couple and co-parenting marital relationship are strategies such as developing common goals, increasing partner respect, securing social support, reducing stress, and instilling hope and service providers and parents of individuals with ASD benefited in receiving information about all the mentioned factors	Canada	The severity of autism as a main diagnosis is considered	TF in the reviewed studies was not considered
Parent and Family Impact of Autism Spectrum Disorders: A Review and Proposed Model for Intervention Evaluation	Karst, and Van Hecke, (2012). [36]	Not Mentioned	Most reviews on ASD intervention considered children as the main focus; parent and family factors are ignored. It is not possible to assume that even significant improvements in the diagnosed child will improve parental distress, especially as the time and expense of intervention might increase family disruption.	USA	Contribution of the different levels of severity of ASD is considered	TF in the reviewed studies is considered
Coping in Parents and Caregivers of Children with Autism Spectrum Disorders (ASD): a Review	Lai and Oei, (2014).) [37]	37 studies	Parental use of coping strategies determined by (1) demographical characteristics (such as gender, age, education, income, and language) and psychological and personal factors (such as personality, cultural values, optimism, sense of coherence, benefit-finding and sense-making abilities, emotional health, and coping styles). It is also concluded that child characteristics (i.e., age, gender, medical conditions, cognitive and adaptive functioning abilities, language difficulties, and behavior problems) and also situational factors (such as treatment availability, family function, and clinician referrals to support resources) are all important determinants.	Singapore	ASD as a general term and main diagnosis	TF in the reviewed studies is considered
A Review of Parent Education Programs for Parents of Children With Autism Spectrum Disorders	Schultz, Schmidt and Stichter (2011) [38]	30 studies	Studies mainly included descriptions of programs for parents of young children with ASD. They are generally focused on a one-on-one training approach. They moderately considered a manual or curriculum. Mostly included data on parent and child outcomes. A majority considered single-case designs to evaluate program affectivity. No data on fidelity of implementation reported in the reviewed studies	USA	The severity of symptoms not mentioned and ASD considered a general diagnostic term.	TF in the reviewed studies was not considered
The Need for More Effective Father Involvement in Early Autism Intervention. A Systematic Review and Recommendations	Flippin and Crais (2011). [39]	27 studies	Considering communication and play as a focal point for the interventions that support fathers’ communication styles and learning needs will likely attract fathers and make them feel more influential in their reciprocity with their child with ASD. Involving fathers effectively in communication and play interventions may reduce maternal stress and boost family cohesion.	USA	ASD has generally used as a diagnostic term	TF in the reviewed studies was not considered
Family-focused autism spectrum disorder research: A review of the utility of family systems approaches	Cridland, Jones, Magee, and Caputi, (2014). [40]	Not mentioned	The theoretical and methodological directions for family-focused ASD research indicates that family systems approaches as a common theoretical framework needs to be more considered in future family-focused ASD research. Considering theoretical concepts such as boundaries, ambiguous loss, resilience, and traumatic growth are all different aspects of family systems TF.	Australia	ASD has generally been used as a diagnostic term	TF in the reviewed studies is considered
Fathers of Youth with Autism Spectrum Disorder: A Systematic Review of the Impact of Fathers’ Involvement on Youth, Families, and Intervention.	Rankin, Paisley, Tomeny, and Eldred (2019). [41]	18 studies	There is a dearth of studies on fathers and ASD. this review suggests that fathers of individuals with ASD play an important role in the life of children with ASD and the family as a whole and should be included in future research on children with ASD.	USA	ASD as a general diagnostic label is considered	TF in the reviewed studies was not considered
Siblings and family environments of persons with autism spectrum disorder: A review of the literature	Smith, and Elder (2010). [42]	12 studies	Factors such as biological, psychological, sociological, and ecological aspects impacted families and siblings are influenced by the context of their families that has already been under the influence of the mentioned factors. To identify people who are at risk of adjustment problem assessment of siblings is necessary.	USA	Autism as a general diagnosis is used	TF in the reviewed studies is considered

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
