# Peer review of "Understanding Different Aspects of Caregiving for Individuals with Autism Spectrum Disorders (ASDs) a Narrative Review of the Literature"

_brainsci, 2020, doi:10.3390/brainsci10080557_

Round 1

Reviewer 1 Report

This is a good attempt by the authors to summarize evidence from published "reviews" that are relevant to informal family caregivers of children with ASD, using the method of "narrative reviews". However here are a few suggestions that will strengthen this paper. Such papers are useful for families, researchers, clinicians, and policymakers. particularly in the region that the authors are in.

Big picture issues

  1. Develop research questions that will be addressed through the articles that will be reviewed (by separating the ideas in the purpose statement).
  2. Operationalize the key phrases for this paper, eg, informal family caregivers or parents of children with ASD. 
  3. Be clear if your review is focused only on children with AASD (below 18 yrs) or aging parents of adult children with ASD.
  4. Develop a culturally relevant review of the literature. Explain how the findings are of relevance to the nations represented by the authors, if the goal to address issues in Iran or UK, or of the immigrant populations in UK?
  5.   A lot of language editing needs to be done (grammar and quality fo language). Pasted below area few specific examples in the next section addressing specific areas of concern.

Specific Issues in the paper

Abstract:

A. To understand associated challenges of caregiving for a child with ASD and to develop the necessary skills and approaches to assist parents of children with Autism Spectrum Disorders (ASD) considerable endeavor has been done. Check language in the latter end of this sentence. Consider rewriting in active voice.

B. Different studies have been stressed the importance and necessitate need for of parental involvement in the intervention process to increase the level of positive impacts.

C. To be able to facilitate parent involvement in intervention implementation which is advisable, the process of caregiving and challenges associated with this process should be understood from different aspects. Consider a clearer rewrite eg. The process of caregiving and the associated challenges should be understood from different aspects to be able to facilitate parent involvement in intervention implementation.

 Introduction: 

A. Suggest that you cite a source to indicate the 70+year history of ASD such as Harris J. Leo Kanner and autism: a 75-year perspective. International Review of Psychiatry. 2018 Jan 2;30(1):3-17.

B. Present Knowledge knowledge on ASD indicates that it is a lifelong neurodevelopmental disorder in which genes play a role but that environmental triggers likely contribute as well

C. There are many single-sentence paragraphs that need to be integrated into bigger paragraphs Eg. on line 34 

D. There are only some affluent countries and on some special occasions in which families given the opportunity of respite care or foster home or residential house [9] This sentence needs to be rewritten in a few separate sentences, case there are several issues being addressed her. Respite care is a break that parents in the affluent countries can have access to which involves a few hours/weekends that the child with special needs will be watched by someone else who will get paid by the state, but it is an underused service for several reasons such as difficulty to find good respite worker. Foster care is a service available in affluent countries like the United States when parents are unable to care for children in their family homes and children have to be removed for a period of time until they are able to be gain parenting skills or improve socio-economic status. Residential care (aka institutionalization) is mostly phased out in most states in USA as disability advocates are moving towards in-home care with supports. Institutional care of children and adults with I/DD are more common in non-affluent countries.   

E. The ultimate goal of this review is to discover the urge of understanding the potential of caregivers and caregiving process for an individual with ASD and to understand different aspects of caregiving for individuals with ASD in different societies and to recognize different impacts that caregiving might have on caregivers

F. Table 1 “Compared to parents of typically developing children or population norms, parents of children with ASD show a poorer quality of life. Contributing factors of parental Quality of life were discovered to be child challenging behavioural behavioral challenges of the child with ASD, parental unemployment, mother caregivers, and lack of social support (of who: Mother?).” Many typos are there in this table. Also, the main findings are hard to find the way the table is presently constructed.

Author Response

Comment:

This is a good attempt by the authors to summarize evidence from published "reviews" that are relevant to informal family caregivers of children with ASD, using the method of "narrative reviews". However here are a few suggestions that will strengthen this paper. Such papers are useful for families, researchers, clinicians, and policymakers. Particularly in the region that the authors are in.

Response:

We appreciate the positive attitude of the reviewer and have done our best to address all the raised issues

Big picture issues

Comment:

Develop research questions that will be addressed through the articles that will be reviewed (by separating the ideas in the purpose statement.

Response:

Three research questions based on the aims of this review added (lines 70 to 760):

This is done to answer the following questions;

Have the presented reviews cover the different aspects of caregiving for an individual with ASD?

How did the presented reviews on caregiving for an individual with ASD echoed different theoretical frameworks to explain the phenomena that are considered to investigate?

How did the presented reviews reveal the geographical distribution of the studies on the impacts of caregiving for an individual with ASD?

Comment:

Operationalize the key phrases for this paper, eg, informal family caregivers or parents of children with ASD.

Response:

We operationalized the key phrase of caregiving of parents of children with ASD in line 120 to 125

The key phrases for this paper are parents of children with ASD and the care that they are providing to their child with ASD. To operationalize this phrase it should be defined as the caregiving and supports characterized by attention to the needs of their child; particularly for those unable to look after themselves sufficiently due to the diagnosis of ASD and involved in the provision of their health or social care.

Comment:

Be clear if your review is focused only on children with AASD (below 18 yrs) or aging parents of adult children with ASD.

Response:

In line 125 we clarified that:

The focus of this narrative review is on the presented reviews on parental caregiving regardless of their age level and it is covered both young and old caregivers.

Comment:

Develop a culturally relevant review of the literature. Explain how the findings are of relevance to the nations represented by the authors, if the goal to address issues in Iran or UK, or of the immigrant populations in UK?

Response:

Based on our understanding of this comment we added the following paragraph (lines 335 to 341)

This is also relevant to high-income countries who admit immigrants from different cultures, to offer more culturally sensitive services concerning the supports provided to parents as caregivers. The main extracted findings from the reviewed reviews in this paper are not unique to the level of the development of the countries and there are particular challenges for parents of these children which is global and of relevance to different nations. Furthermore, additional cross-cultural research, albeit within a local context in different countries, is essential if the international understanding of ASD is to be boosted globally.

Comment:

 A lot of language editing needs to be done (grammar and quality fo language). Pasted below area few specific examples in the next section addressing specific areas of concern.

Response:

Due to the shortage of the feedback time (5 days ) we did an online English Language system and made some corrections

Specific Issues in the paper

Abstract:

  1. To understand associated challenges of caregiving for a child with ASD and to develop the necessary skills and approaches to assist parents of children with Autism Spectrum Disorders (ASD) considerable endeavor has been done. Check language in the latter end of this sentence. Consider rewriting in active voice.

Response:

The sentence was rewritten.

  1. Different studies have been stressed the importance and necessitate need for of parental involvement in the intervention process to increase the level of positive impacts.

Response:

The sentence was rewritten.

  1. To be able to facilitate parent involvement in intervention implementation which is advisable, the process of caregiving and challenges associated with this process should be understood from different aspects. Consider a clearer rewrite eg. The process of caregiving and the associated challenges should be understood from different aspects to be able to facilitate parent involvement in intervention implementation.

Response:

The sentence was rewritten.

 Introduction:

  1. Suggest that you cite a source to indicate the 70+year history of ASD such as Harris J. Leo Kanner and autism: a 75-year perspective. International Review of Psychiatry. 2018 Jan 2;30(1):3-17.

Response:

We really appreciate this very useful comment and suggestion and updated this refernce

  1. Present Knowledge knowledge on ASD indicates that it is a lifelong neurodevelopmental disorder in which genes play a role but that environmental triggers likely contribute as well

Response:

The sentence was rewritten.

  1. There are many single-sentence paragraphs that need to be integrated into bigger paragraphs Eg. on line 34

  1. There are only some affluent countries and on some special occasions in which families given the opportunity of respite care or foster home or residential house [9] This sentence needs to be rewritten in a few separate sentences, case there are several issues being addressed her. Respite care is a break that parents in the affluent countries can have access to which involves a few hours/weekends that the child with special needs will be watched by someone else who will get paid by the state, but it is an underused service for several reasons such as difficulty to find good respite worker. Foster care is a service available in affluent countries like the United States when parents are unable to care for children in their family homes and children have to be removed for a period of time until they are able to be gain parenting skills or improve socio-economic status. Residential care (aka institutionalization) is mostly phased out in most states in USA as disability advocates are moving towards in-home care with supports. Institutional care of children and adults with I/DD are more common in non-affluent countries.

Response:

We appreciate this very useful comment and added this to the proposed part.

  1. The ultimate goal of this review is to discover the urge of understanding the potential of caregivers and caregiving process for an individual with ASD and to understand different aspects of caregiving for individuals with ASD in different societies and to recognize different impacts that caregiving might have on caregivers

Response:

The sentence was rewritten.

  1. Table 1 “Compared to parents of typically developing children or population norms, parents of children with ASD show a poorer quality of life. Contributing factors of parental Quality of life were discovered to be child challenging behavioural behavioral challenges of the child with ASD, parental unemployment, mother caregivers, and lack of social support (of who: Mother?).” Many typos are there in this table. Also, the main findings are hard to find the way the table is presently constructed.

Response:

The sentence was rewritten and the format of the table in accordance to the guideline of the Brain Science was re-ordered.

Reviewer 2 Report

Dear Authors,

thank you for the opportunity to review the paper titled 'Understanding Different Aspects of Caregiving for Individuals with Autism Spectrum Disorders (ASD) A NARRATIVE REVIEW OF THE LITERATURE'

ABSTRACT 

It is clear, informative and well structured

INTRODUCTION

It is well written and the references are well cited

MATERIALS AND METHOD

I appreciated the work you have done; table is clear and complete

DISCUSSION

It is complete

Author Response

We appreciated the very positive comments  you kindly provided.